# Dam Crack Image Detection Model on Feature Enhancement and Attention Mechanism

**Guoyan Xu \*, Xu Han, Yuwei Zhang and Chunyan Wu**

College of Computer and Informatica, Hohai University, Nanjing 210094, China
* Correspondence: gy_xu@126.com

**Abstract:** Dam crack detection can effectively avoid safety accidents of dams. To solve the problem that the dam crack image samples are not available and the traditional algorithm detects cracks with low accuracy, we provide a dam crack image detection model based on crack feature enhancement and attention mechanism. Firstly, we expand the dam crack image dataset through a generative adversarial network based on crack feature enhancement (Cracks Enhancements GAN, CE-GAN). It can fully expand the dam crack data samples and improve the quality of the training data. Secondly, we propose a crack image detection model based on the attention mechanism (Attention-based Faster-RCNN, AF-RCNN). The attention mechanism is added in the crack detection module to give different weights to the proposal boxes around the crack target and fuse the candidate boxes with high weights to accurately detect the crack target location. The experimental results show that our algorithm achieves 81.07% mAP on the expanded dam crack dataset, which is 8.39% higher than the original Faster-RCNN algorithm. The detection accuracy is significantly improved compared with other traditional dam crack detection algorithm models.

**Keywords:** dam crack detection; FSL; GAN; attention mechanism; feature enhancement

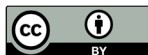

## 1. Introduction

China has nearly 100,000 reservoir dams, making it the country with the most reservoir dams in the world. Reservoir dams have functions such as flood control, power generation, water supply and irrigation, which can protect people's property and also promote economic development. In general, China's reservoir dams show a wide distribution, the total number of large spatial differences, many types of dams, and other characteristics, so doing a good job of reservoir dam safety inspection work is of great significance, although the task is heavy. However, with the increasing of the service life of dams, the aging of materials, environmental corrosion, and other reasons, the surface and interior of the dams are inevitably deformed and cracks of varying degrees appear. For example, after more than half a century of operation of the Huai River tributary of the Foziling Reservoir dam, fork joints and leakage occurred at the junction of the construction joints and deformation joints at the arch ring of the dam [1]. Another example is the dam on the lake reservoir located in Baishan City, Jilin Province, which has multiple cracks at the top, 30–80 cm wide and nearly 2 m deep, causing particularly severe leakage and deformation damage to the dam. As there are a large number of dams in China, there are many such examples. If these cracks are not detected in time, they will have a negative impact on the safe operation of the dam. The current method for dam crack detection is based on manual inspection, which is very labor-intensive, and the accuracy of the detection depends on the professional level and experience of the inspector. Since different inspectors have different results in determining cracks, there is a subjective error in judgment. In addition, the manual inspection method has potential safety problems for the

inspectors, and it is not easy to collect a large amount of relevant data. Therefore, it is hardly possible to satisfy the needs of dam crack detection using manual inspection methods alone, and it has become a priority to explore an objective, safe, and accurate dam crack detection method. With the development and application of deep learning, deep learning dam crack detection algorithms have gradually replaced manual detection. By applying deep learning to crack detection, Zheng et al. [2] have made theoretical basis and practical results for crack detection on the surface of buildings such as roads, bridges, houses, and dams based on deep learning. Ali et al. [3] introduced the important research of using convolutional neural networks (CNN) to classify and segment crack images to detect structural cracks, which can help us better understand the application of deep learning in crack detection. Although many researchers have studied crack detection based on deep learning, there are still some problems in dam crack detection due to the particularity of crack images in dam crack detection. Dam crack detection mainly faces two problems at the present. The first problem is the difficulty and high cost of obtaining images of cracks in dams, so it is very hard to have a large sample dataset. The second problem is that the speed and accuracy of traditional dam crack detection models are low, and the mistakes are large.

To address the first problem in dam crack detection, researchers usually apply data augmentation to address the problem of insufficient data size, i.e., data augmentation and feature enhancement of small sample sets with the help of auxiliary information. Researchers now typically use generative adversarial networks (GAN) [4] to solve low data size problems. GAN can generate clear and different images from the original images. Researchers can achieve a high-quality extension of the dam crack image set by using an improved adversarial generative network. To address the second problem in dam crack detection, researchers use deep learning to study dam crack images, which can effectively improve the speed and accuracy of dam crack detection. The deep learning crack detection method, which combines spatial and temporal pattern mining of crack features and deep convolutional neural networks, achieves accurate localization of cracks and improves crack detection accuracy.

Given the deficiencies of the above models, we provide a crack image detection model with feature enhancement and attention mechanism. Our main contributions are as follows:

(1) We propose a generative adversarial network based on crack feature augmentation (Cracks Enhancements GAN, CE-GAN) to expand the dam crack dataset. We add the image crack enhancement module to the model so that CE-GAN can better learn the features of dam cracks and generate crack samples that are closer to the crack features of real samples to satisfy the demand of expanding data samples.

(2) We propose a crack image detection model based on the attention mechanism (Attention-based Faster-RCNN, AF-RCNN) to facilitate the improvement of detection accuracy. We introduce an attention mechanism to give different weights to the proposal boxes around the target and use weighted summation to fuse the selected proposal boxes, so that the updated candidate proposal box is the optimal one and the feature vector of the candidate proposal box contains more accurate location information.

(3) We improve the base anchor aspect ratio generated by the AF-RCNN based on the statistics of crack aspect ratio in the original dam crack image, so that the adjusted anchor is more suitable for crack detection and improves the efficiency of crack localization.

The rest of this paper is organized as follows. Section 2 reviews related works and discusses their limitations. Section 3 describes the framework of the dam crack detection model. Section 4 verifies the effectiveness of the proposed model through comparative experiments. Section 5 provides a relevant discussion of the deficiencies and developable directions demonstrated by the experimental results. Section 6 offers some conclusions and suggestions for future work.

## 2. Related Work

### 2.1. Generative Adversarial Network in Crack Image Generation

The generative adversarial network was a generative model proposed by Goodfellow et al. [4] in 2014, which differs from the traditional generative model in that the model structure contains a generative part and an adversarial part. In 2015, Radford et al. [5] proposed the DCGAN (deep convolutional GAN) model. This is the first time a convolutional neural network has been applied to a generative adversarial model. In 2017, Isola et al. [6] proposed the Pix2Pix model, which realized the mapping from input image to output image. Its training process no longer requires relevant matched pairs of image information, thus simplifying the process of preliminary data preparation and providing technical support for further subsequent applications.

There have been many applications of GAN in the generation of crack images. Hu Min et al. [7] applied GAN to the generation of crack images. Their experiments proved that some results were achieved for crack images, but this method was very prone to the problem of training instability, which led to a single pattern in the generated images. To solve the problem of training instability, Arjovsky et al. [8] used Wasserstein distance to measure the difference between real data and generated data, but the weight cropping of this method limits the performance of the network. Gulrajani et al. [9] showed that the use of gradient penalty term instead of weight clipping improves the defect of poor network performance. Radford et al. [5] modified the structure of the generative and discriminative models of GAN to use deep convolutional neural networks into GAN, and then they proposed deep convolutional generative adversarial networks (DCGAN). Wei et al. [10] applied DCGAN to the generation of crack images. They demonstrated that the model could obtain higher quality image samples through experiments, but they found that the generated image details are not clear enough and the generated images are affected by the number of training samples.

From the current research, it can be seen that the research on crack image generation is mainly focused on pavement, bridge, and tunnel cracks. There are still relatively few research studies on dam crack image generation. However, the background of dam fracture images is rather cluttered, which tends to lead to a lack of clear details in the generated fracture images.

### 2.2. Deep Learning in Crack Target Detection

With the development of deep learning and neural networks, crack detection based on deep learning has attracted more and more attention. Han et al. [11] applied the deep convolution neural network to the detection of asphalt pavement cracks, which provided an alternative solution for the automatic detection of pavement cracks. Mohammed et al. [12] evaluated and verified three commonly used crack detection models of concrete structures, and expounded the advantages and disadvantages of each model. Weng et al. [13] proposed a segmentation method based on an improved full convolutional neural network in order to complete the segmentation and detection of pavement cracks in complex environments, which can segment the pavement cracks more accurately. Zhu et al. [14] used multilayer convolution to automatically extract crack features. They achieved the integration of local and abstract features of cracks using the superposition of shallow and deep networks, which preserved the crack detail features. To improve the detection accuracy of cracks, Sun et al. [15] proposed crack image recognition based on Faster-RCNN, which achieved certain results for the detection and accurate localization of image cracks, but there are more leakage cases for the detection of smaller cracks. Xue et al. [16] implemented the detection of crack defects based on the improved Faster-RCNN, with backbone using the improved inception full convolutional network to obtain the feature map and improve the feature extraction capability. He et al. [17] combined multi-level feature maps in backbone into multi-scale feature maps to obtain more contexts feature information. However, both of them suffer from the problem that the candidate proposal box

is not accurate enough for locating the target cracks. Ding et al. [18] improved the localization accuracy by setting different proportions of anchor frames through mean clustering, which made the network adapt to the localization of smaller cracks. However, the impact of the non-maximum suppression (NMS) algorithm on the candidate suggestion window leads to inaccurate target detection localization, resulting in the problem of target crack misdetection as well as missed detection. Li et al. [19] proposed a new application scenario for applying YOLOv3 [20] to crack detection in floodgate dam surface and shared its effects. YOLOv3 uses three scale feature maps for prediction and enhances the detection of small cracks. Feng et al. [21] proposed a method of crack detection on dam surface (CDDS) using deep convolution network, and the CDDS network is improved based on the characteristics of the SegNet [22] structure and consists of encoding and decoding parts. Chen et al. [23] designed a shallow encoding network to extract features of crack images based on the statistical analysis of cracks. Furthermore, to enhance the relevance of contextual information, they introduced an attention module into the decoding network.

From the current research, it can be seen that there is still much space for improvement in the dam crack detection algorithm.

## 3. The Dam Crack Image Detection Model

The overall framework of the dam crack image detection model is shown in Figure 1. Compared with the traditional dam crack detection algorithm, the model provides innovations in image expansion and crack detection work.

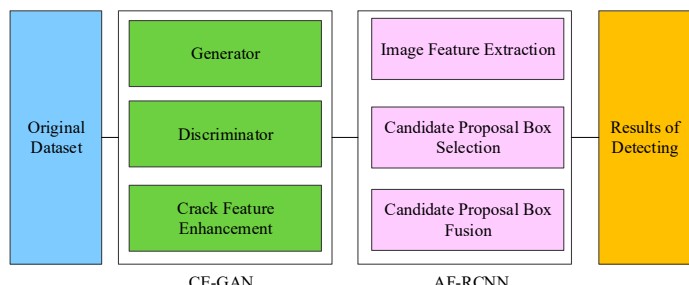

**Figure 1.** The framework of our model. Compared with the traditional dam crack detection algorithm, our model provides innovations in image expansion and crack detection. When the dam crack dataset is fed into the model, the expansion module effectively expands the dataset and enhances the crack feature through generator and discriminator, so as to facilitate the training of the dam crack detection network. The crack detection module extracts the feature under the action of the attention mechanism, generates candidate proposal boxes, and optimizes them. Finally, the model generates detection results.

### 3.1. Generative Adversarial Network Based on Crack Feature Enhancement

To address the problems of fewer dam crack images and more cluttered image background, we want to build a network model that is more conducive to dam crack image generation. Therefore, we propose the Cracks Enhancements GAN (CE-GAN), which is a generative adversarial network based on crack feature enhancement.

CE-GAN applies the self-encoder and self-decoder to the discriminator, which can better distinguish the real crack image from the generated crack image. The generator uses a deconvolutional neural network, which can reduce the feature distance and decrease the value of the loss function. This model incorporates an image crack enhancement module to better learn the features of dam cracks, further reduce the feature differences between the generated crack images and the real crack images, as well as improve the quality of the generated images. The structure of the model is shown in Figure 2.

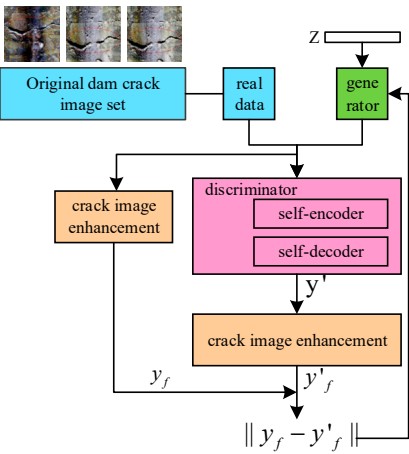

**Figure 2.** The architecture of CE-GAN.

We define $x$ as the real dam crack image, $Z$ as the random noise data input to the generative model, and $G(z)$ as the generative model that generates fake images based on the noise data. $G(z)$ and $x$ can generate the image $y_f$ by the crack feature enhancement module. In the other channel of the model, the real image $x$ is fed into the discriminator for training. The trained discriminator can be obtained after a period of training, which enables the discriminator to learn the features of the cracked image. After this, $G(z)$ and $x$ are fed into the trained discriminator to obtain $y'$. $y'$ can generate the image $y'_f$ by the crack feature enhancement module. Then, we calculate the characteristic values of the image $y_f$ and the image $y'_f$ and subtract the two feature values to obtain the difference which is passed to the generator through the feedback channel. If the image reconstructed by the discriminator is close to the real image, the difference of the characteristic values will be relatively small. Similarly, if the difference between the reconstructed image of the discriminator and the real image is relatively large, the characteristic values difference will also be relatively large, resulting in a large loss error, thus prompting the generator to update the parameters so that the output crack image can be closer to the real crack image.

### 3.1.1. Crack Enhancement

By analyzing the collected dam crack images in the dataset, we found some shortcomings about dam crack images and their data sets:

(1) The semantics of dam crack images are relatively simple. Dam crack images need to focus on the underlying features, such as the outline, color, texture, and morphology of the cracks. As shown in Figure 3a, the crack profiles are mostly straight and the texture of the crack images is not clear enough. According to this feature, in the image expansion part, we could design filters to filter the image data by Fourier transform to achieve the enhancement of the crack image.

(2) The dam cracks have various characteristics and shapes. The formation of dam cracks is affected by internal and external uncertainties, resulting in a cluttered background of the crack image. The crack image has obvious linear characteristics. There are also multiple fracture branches near the fracture, as shown in Figure 3b,c). According to this feature, in the crack detection part, we could set the anchor frame suitable for dam crack detection by calculating the aspect ratio of linear cracks.

(3) The dam crack sample set is lacking. The sample set was acquired mainly from the dam crack images collected from a dam safety inspection report, but the amount of data in the safety inspection report that met the experimental requirements was relatively small, resulting in insufficient data for subsequent model training.

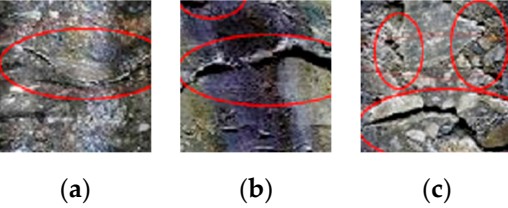

(**a**)    (**b**)    (**c**)

**Figure 3.** Sample images of dam cracks. (**a**) shows the image where the crack is not obvious. (**b**) shows the image where the crack is obvious. (**c**) shows the image with multiple cracks.

In order to solve the above defects, in this paper, we use the Fourier transform in the crack feature enhancement module to convert the input image from the spatial domain to the frequency domain and filter the image t      o enhance the image features of the dam crack, so that the model can better learn the features of the dam crack and make the generated crack sample closer to the crack image of the real sample.

The two-dimensional discrete Fourier transform [24] transforms the image from the spatial domain to the frequency domain. The transformation equation is shown in Equation (1).

$$F(u,v) = \sum_{x=0}^{M-1}\sum_{y=0}^{N-1} f(x,y)e^{-j2\pi(\frac{ux}{M}+\frac{vy}{N})} \tag{1}$$

where M is the image length and N is the image height. Here, $F(u,v)$ denotes a frequency domain image, where the range of $u$ is [0, M − 1] and the range of $v$ is [0, N − 1]. $f(x,y)$ denotes the spatial domain image, a spatial domain matrix of size $M \times N$, where the range of $x$ is [0, M − 1] and the range of $y$ is [0, N − 1].

Equation (2) is the transfer function of the Butterworth high-pass filter.

$$H(u,v) = \frac{1}{(1+[\frac{D_0}{D(u,v)}]^{2n})} \tag{2}$$

where $D_0$ is the specified positive number, $D(u,v)$ denotes a distance from the point $(u,v)$ to the center of the filter, and $n$ is the order of the filter. If $H_{HP}(u,v)$ is the transfer function of the high-pass filter, then the transfer function of the corresponding low-pass filter is $H_{LP}(u,v) = 1 - H_{HP}(u,v)$.

We use the high-pass filter as the baseline filter to improve the dam crack image by controlling the low-frequency information and adjusting the high-frequency information accordingly, as shown in Equation (3).

$$g(x,y) = f^{-1}[k_1 F(u,v) + k_2 H_{HP}(u,v)F(u,v)] \tag{3}$$

where $F(u,v)$ denotes a frequency domain image which is obtained from the input image $f(x,y)$ by Fourier transform, $H_{HP}(u,v)$ denotes the high-pass filter, $k_1$ is the variable that controls the low frequency, and $k_2$ is the variable that controls the high frequency.

We found by testing that the image enhancement is better when $k_1 = 1$ and $k_2 = 1.3$. We perform low-pass filtering, high-pass filtering, and improved filter filtering operations on the image, respectively. The effect is shown in Figure 4.

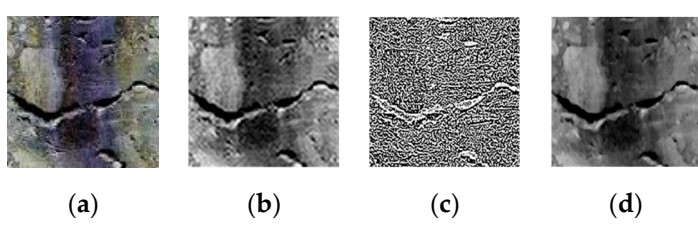

(**a**)    (**b**)    (**c**)    (**d**)

**Figure 4.** The effect of three kinds of filtering methods. (**a**) shows the original dam crack image. (**b**) shows the effect under low-pass filtering. (**c**) shows the effect under high-pass filtering. (**d**) shows the effect under our improved filtering.

As shown in Figure 4a, the crack edges in the high-frequency region and the surrounding area become blurred when the dam crack image is processed with low-pass filtering, because the crack edge information in the high-frequency region is filtered out when the low-pass filtering filters the high-frequency information. As shown in Figure 4b, after high-pass filtering, most of the background in the original image will be lost. Although the high-pass filter can enhance the crack features, the loss of background is also detrimental to the subsequent detection of cracks in the dam. As shown in Figure 4c, the improved filter enhances the crack features of the image, and the background of the image is not lost. In summary, this image data enhancement method can achieve the enhancement of dam crack image features.

3.1.2. Generator and Discrimination

The input of the generator model is a noisy data Z that obeys some random distribution. The output of the generator is a new sample of size 256 × 256 × 1. The generator has no fully connected layers and pooling layers. After a reshape layer, it will go through five deconvolutional layers, with a convolution kernel size of 5 × 5. The specific structure is shown in Figure 5.

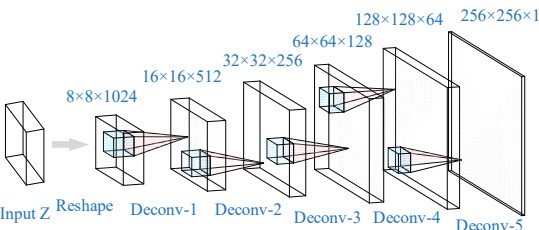

**Figure 5.** The architecture of the generator.

The discriminator consists of an encoder and a decoder. The specific structure is shown in Figure 6.

The encoder consists of three convolutional layers and three pooling layers. The input to the encoder is an image of size 256 × 256 × 1 generated by the generator. After transforming the number of channels and image size for the feature map, the dimension of the input data is compressed to 32 × 32 × 32. All the convolutional layers in the encoder use the ReLU activation function. The size of the convolution kernel is 3 × 3.

The decoder consists of four convolutional layers and three up-sampling layers. The input data of the decoder is the output data of the encoder. The output of this network is a new image sample data of 256 × 256 × 1. The size of the convolution kernel is 3 × 3.

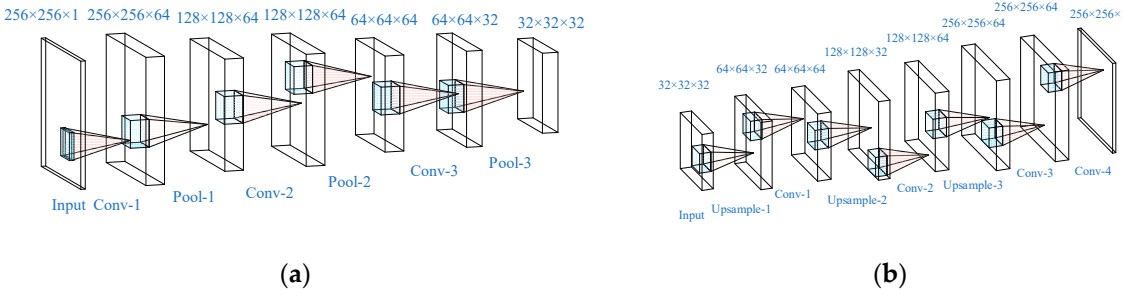

(**a**)                                                                (**b**)

**Figure 6.** The architecture of discriminator. (**a**) is the structure of encoder. (**b**) is the structure of decoder.

### 3.1.3. Loss Function

The loss function of the original generative adversarial network is shown in Equation (4). We adjust the loss function to make the image generation stable and diverse.

$$\min_G \max_D V(D,G) = E_{x \sim P_{data}}(x) \log[D(x)] + E_{z \sim P_z} \log[(1 - D(G(z)))] \tag{4}$$

Set $I$ is the image, and we find the eigenvalues of images $y_f$ and $y'_f$ to obtain $A(y_f)$ and $A(y'_f)$. We obtain the two-parametric number of eigen differences $||A(y_f) - A(y'_f)||_2$ and feed the feature difference to the generator as part of the loss function, as shown in Equation (5).

$$L_G(z) = D(G(z)) = \frac{(\dfrac{|| A(y_f) - A(y'_f) ||_2}{i + j})}{256} \tag{5}$$

where $i$ is the image of the row pixel size and $j$ is the image of the column pixel size.

The loss function of the discriminator is shown in Equations (6)–(8).

$$L_D(x,z) = D(x) + (-D(G(z))) \tag{6}$$

$$D(x) = \frac{(\dfrac{|| A(y_f)_x - A(y'_f)_x ||_2}{i + j})}{256} \tag{7}$$

$$D(G(z)) = \frac{(\dfrac{|| A(y_f)_{G(z)} - A(y'_f)_{G(z)} ||_2}{i + j})}{256} \tag{8}$$

As shown in Equation (9), we add a threshold $M$ to the discriminator loss function thus acting as a restriction to avoid model collapse and to balance $D(x)$ and $D(G(z))$.

$$L_D(x,z) = D(x) + \max(0, M - D(G(z))) \tag{9}$$

Finally, we improve the loss function as shown in Equations (10) and (11).

$$f_{PT}(S) = \gamma \sum_{i,j,i \neq j} \cos(e_i, e_j) \tag{10}$$

$$L_D(x,z) = D(x) + \max(0, M - D(G(z))) + f_{PT}(S) \tag{11}$$

where $\gamma$ is hyperparameters, $e_i$ denotes the output vector after decoding in a batch S=$\{..x_i..x_j..\}$, and $f_{PT}(S)$ is used to represent the similarity of the images.

### 3.2. Crack Image Detection Based on Attention Mechanism

To address the problems of low accuracy of traditional detection algorithms and inaccurate positioning of candidate suggestion frames in the process of detecting dam crack images, we propose the AF-RCNN, a crack image detection model based on the attention mechanism. This model uses the Faster-RCNN model as the baseline model. Its feature extraction network partly combines ResNet-50 [25] and SENet [26] to improve the feature representation capability and quality of the network. It adjusts the scale and proportion of the foundation anchor according to the dam crack aspect ratio. In addition, the model achieves candidate proposal boxes merging by using the attention mechanism to improve the accuracy of the target proposal boxes for crack localization, reduce the occurrence of

false detection and missed detection, and promote the improvement of the detection accuracy. The structure of the model is shown in Figure 7a.

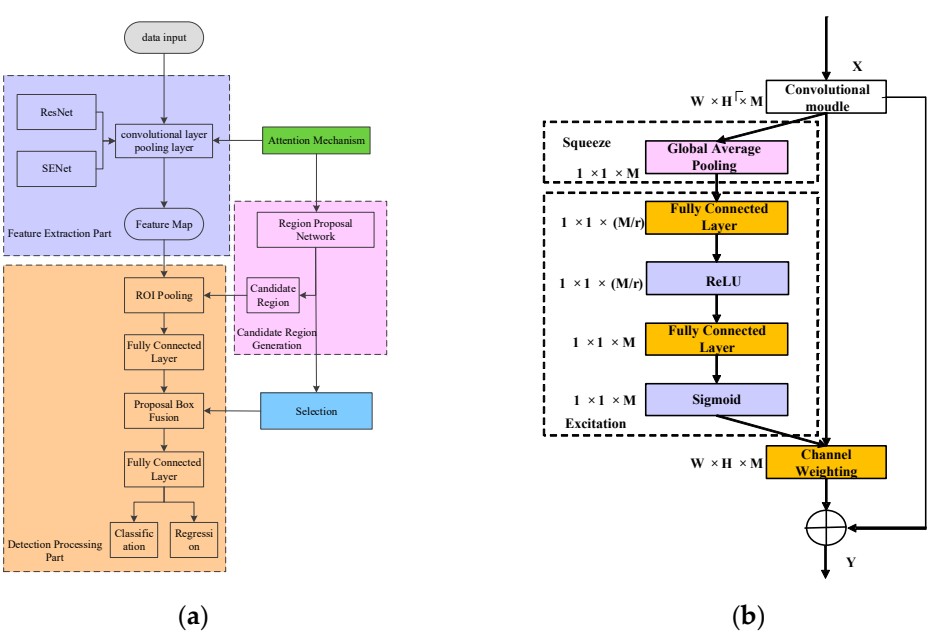

(**a**)                                                          (**b**)

**Figure 7.** (**a**) Structure of crack image detection model based on attention mechanism. (**b**) Residual unit embedded SENet.

### 3.2.1. Feature Extraction Network with SENet Structure

Considering that the network depth is not easily too deep, we use the ResNet-50 network as the benchmark network. The core operation of the SENet network consists of two parts: squeeze and excitation. We try to embed the SE structure into other network models and find that this enables the network model to integrate more features spatially, allowing the network to focus more on the feature channels with more information while suppressing the feature channels with less information, thus improving the feature representation capability of the network. As a result, we embed the SENet structure in the feature extraction network Resnet-50 in AF-RCNN, so that it can incorporate more spatial features, perform feature compression for feature channels with different information amounts, obtain corresponding different weights after squeezing the features, and update the relevant channel information according to the different weights.

The structure of the residual unit added to the SENet network is shown in Figure 7b, where X is a feature map and Y is a feature map after squeeze and excitation. First, input the feature map of size $M \times W \times X$ into the SENet structure, perform global average pooling on the feature map, and squeeze it into a vector of size $M \times 1 \times 1$; second, input the real number sequence of size $M \times 1 \times 1$ into the fully connected layer, reduce the dimension into a vector of size $1 \times 1 \times (M/r)$, and increase the dimension to a vector of size $1 \times 1 \times M$ through the ReLU activation function; third, input the vector of size $1 \times 1 \times M$ into the second fully connected layer and obtain the corresponding weight of the channel through the sigmoid activation function; and finally, calculate the channel and the corresponding weight to realize the channel weighting and obtain the updated output.

We embed the SENet structure in the feature extraction network Resnet-50, i.e., SE units are embedded after the Conv1, Conv2_3, Conv3_4, Conv4_6, and Conv5_3 layers, respectively. The specific structure is shown in Figure 8.

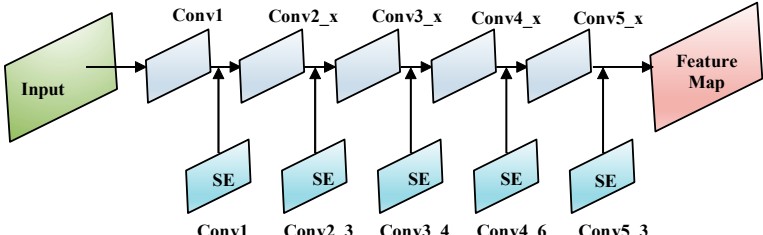

**Figure 8.** Resnet-50 embedded SENet**.**

3.2.2. Improvement of Regional Proposal Network

We calculate the length and width of the crack using the following equation, and then set the size of the anchor applicable to dam crack detection based on the crack aspect ratio. The calculation of the continuous curve spacing [27] is shown in Figure 9. For two continuous functions $U(x)$ and $V(x)$, $(U(x) + V(x))/2$ denotes the midpoint connection of the functions, $P(x) = U(x) - V(x)$ denotes the vertical distance, and $\theta(x) = [(U(x) + V(x))/2]'$ denotes the inclination of the curve at the midpoint $P$. So, $W(x) = P(x) \times cos(\theta(x))$ is an approximation of the width between the two curves where $P$ is located.

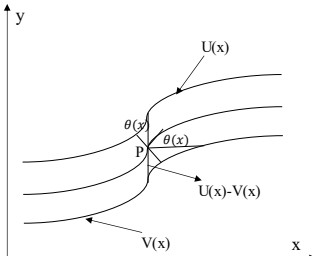

**Figure 9.** Diagram of continuous curve spacing.

Based on the above calculation, the length and width of the dam cracks can be calculated using the following steps:

(1) Transverse cracks

Step 1: Average the position coordinates of all crack pixel points in column *j* as shown in Equation (12).

$$M(j) = \frac{\sum_{k=1}^{P(i)} z(k, j)}{P(j)} \tag{12}$$

where $P(j)$ denotes the number of pixels in column $j$ and $Z(k, j)$ denotes the position coordinates of all crack pixels in the image, $k$ = 1, 2, …, $P(j)$.

We can consider the line between the coordinates $(M(j), j)$ as the centerline of the crack, and then the inclination angle of the centerline of the *j*th column is expressed as Equation (13).

$$\theta(j) = \arctan[\frac{(M(j+r) - M(j-r))}{(2 \times r)}] \tag{13}$$

where $r = 1, 2, 3$.

We take the *j* column as the center, calculate the inclination angles of the left and right columns of the *j* column, and then take the average $\overline{\theta}(j)$ of the three columns as the inclination angle of the *j*th column.

Step 2: Calculate the width of each column where the crack is located separately. $W(j)$ denotes the crack width in column *j*, as shown in Equation (14).

$$W(j) = \mu \times P(j) \times \cos(\overline{\theta}(j)) \tag{14}$$

where $\mu$ is a pixel resolution, $P(j)$ denotes the number of pixels in column *j*, and $\overline{\theta}(j)$ is the inclination of the *j*th column.

Step 3: Calculate the length of cracks, as shown in Equation (15).

$$L = \frac{\mu \times \sum\limits_{i=1}^{m} P(j)}{\dfrac{\sum\limits_{j=1}^{m} P(j) \times \cos(\overline{\theta}(j))}{n}} \tag{15}$$

(2) Longitudinal cracks are calculated as above, as shown in Equations (16) and (17).

$$W(i) = \mu \times P(i) \times \cos(\overline{\theta}(i)) \tag{16}$$

$$L = \frac{\mu \times \sum\limits_{i=1}^{m} P(i)}{\dfrac{\sum\limits_{i=1}^{m} P(i) \times \cos(\overline{\theta}(i))}{m}} \tag{17}$$

where $P(i)$ denotes the number of pixels in row *i*.

We use the above formula to calculate the length and width of the crack, and then set the size of the anchor applicable to dam crack detection based on the aspect ratio of the crack.

### 3.2.3. Proposal Boxes Fusion with Attention Mechanism

According to the problems of the NMS algorithm, in order to obtain more information about the location of the target crack, the suppressed surrounding candidate proposal boxes need to be selected. We need to select the surrounding candidate proposal boxes that contain more target location information. However, different surrounding proposal boxes contain different location information and thus have different importance for the target proposal box. We propose a weighted fusion method of candidate proposal boxes based on the attention mechanism. Firstly, we will set a reasonable threshold T to select the surrounding proposal boxes. Secondly, we will identify the surrounding proposal boxes as the one that contains more target locations when the score of intersection over union (IOU) between the target proposal box and the surrounding proposal box is greater than or equal to T. Finally, we fuse the selected proposal boxes using multiplexing and weighted summation to achieve the updated target proposal frame as the optimal target proposal box, so that the feature vector of the target proposal box contains more accurate location information and improves the accuracy of model recognition. The specific screening process is shown in Figure 10.

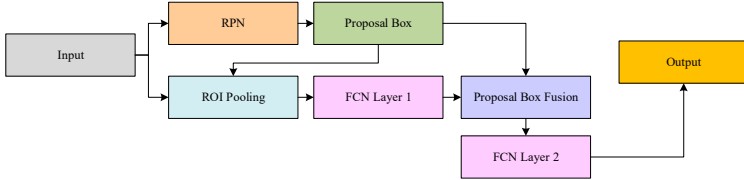

**Figure 10.** Proposal boxes fusion.

The steps to achieve the fusion of surrounding proposal boxes based on the attention mechanism are as follows:

Step 1: We input the RPN selected target proposal box to the fully connected layer 1.

Step 2: $P = [p_1, p_2, \ldots, p_i]$ denotes the feature vector on the fully connected layer corresponding to the target proposal box. We identify the surrounding proposal boxes as the one that contains more target locations when the score of intersection over union (IOU) between the target proposal box and the surrounding proposal box is greater than or equal to *T*. After that, we fuse the selected proposal boxes by using multiplexing and weighted summation to improve the feature vector of the target proposal box P.

Step 3: We calculate the weights between the target proposal box P and the retained feature vectors of each surrounding proposal box. Then, we combine the attention mechanism to automatically learn the weight information, as shown in Equation (18). We use the Softmax classification to output the weights.

$$e_{ij} = W^T(W_a[f_{v_i}, f_{v_j}]) \tag{18}$$

where $W^T$ and $W_a$ denotes the parameters of automatic learning and $f_{v_i}$ and $f_{v_j}$ denotes the feature vectors on the fully connected layer 1, which correspond to the *i*th and *j*th surrounding proposal boxes of the target proposal box vector.

Step 4: We update the feature vector corresponding to the target proposal box P, as shown in Equation (19).

$$F_{V_i} = a_{i1} \times F_{V_1} + a_{i2} \times F_{V_2} + \ldots + a_{ij} \times F_{V_j} \tag{19}$$

Step 5: We output the updated target proposal box to the fully connected layer 2.

## 4. Experimental Result and Analysis

### 4.1. CE-GAN Experiment Results and Analysis

The crack dataset was obtained from the inspection results of a power station dam project. The original dataset has a total of 759 images. The crack dataset is adjusted after collection so that the target area of the crack is basically located in the image. In this experiment, the resolution of the crack image is uniformly modified to 256 × 256 size, as shown in Figure 11.

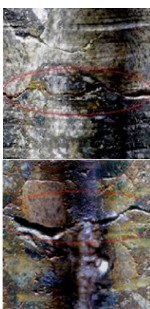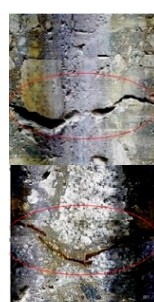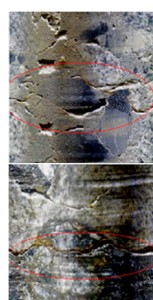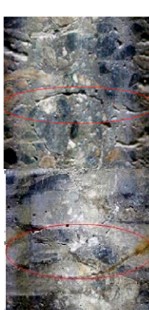

**Figure 11.** The samples of the dataset.

In this experiment, we train the generator and discriminator using a set of dam crack images, both of size 256 × 256. The initial weights are set to follow a normal distribution of N (0, 0.02). We compare the experimental results with those of other models to verify the effectiveness of the model. Among the selected comparison models are GAN [4], WGAN [8], WGAN-GP [9], and DCGAN [10].

In order to visually compare the quality of images generated by different generative adversarial networks and their visual effects, images generated by the models GAN, WGAN, WGAN-GP, DCGAN, and CE-GAN are shown in Figure 12.

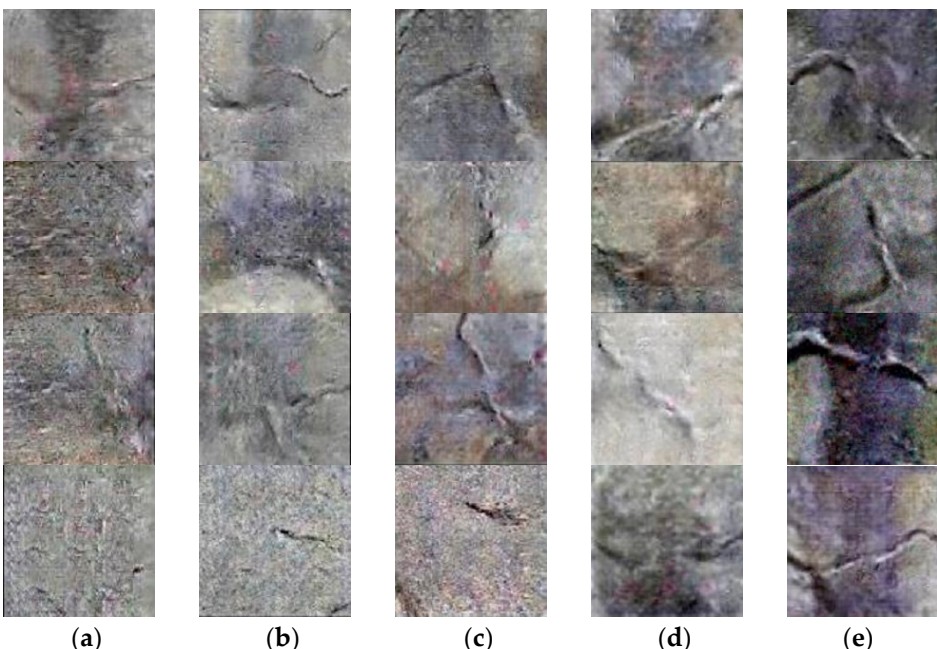

|   (**a**)   |   (**b**)   |   (**c**)   |   (**d**)   |   (**e**)   |

**Figure 12.** Crack images generated by different models. (**a**) Crack images generated by GAN. (**b**) Crack images generated by WGAN. (**c**) Crack images generated by WGAN-GP. (**d**) Crack images generated by DCGAN. (**e**) Crack images generated by CEGAN.

We found that the images generated by the GAN basically have no crack features and there is a pattern collapse, which cannot meet the required sample set. The images generated by the WGAN and DCGAN appear to have crack features, but the details of the crack images are missing, and they cannot learn the features of the dam cracks well, and the diversity of the generated images also performs poorly. Although the image crack features generated by the WGAN-GP are already clearer and better than WGAN and DCGAN in image quality, the diversity of the generated images as well as the image quality are inferior to CE-GAN. The dam crack images generated by the CE-GAN are closer to the real dam crack images and are the best among the generated images in terms of visual effects.

To further compare the model effect, we selected FID evaluation index to analyze and compare the models of WGAN, WGAN-GP, DCGAN, and CE-GAN. FID is an index to evaluate the generative model, which responds to the difference between the generated image distribution and the original image distribution; the smaller the FID value is, the smaller the difference is and the more realistic the generated image is. The formula is as shown in Equation (20).

$$FID = \| \mu_r - \mu_g \|^2 + Tr(\Sigma_r + \sum_g - 2(\Sigma_r \sum_g)^{\frac{1}{2}}) \tag{20}$$

where $r$ is the real image, $g$ is the generated image, $\mu$ is the mean of the real image features, $\Sigma$ is the covariance, and $Tr$ is the trace of the matrix.

As shown in Figure 13, it can be seen that CE-GAN converges faster than WGAN, DCGAN, and WGAN-GP in the early stage. The FID value of CE-GAN is also lower than the other three models. A smaller FID value indicates a higher quality of the generated image. So, CE-GAN is useful for the improvement of dam crack images.

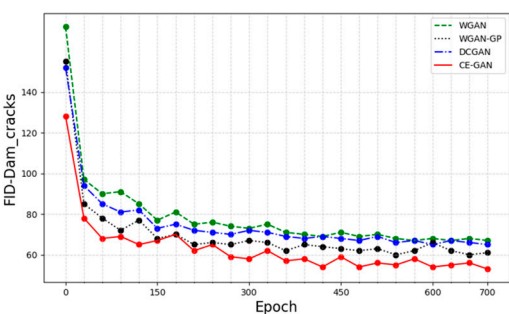

**Figure 13.** The graph of FID value.

### 4.2. AF-RCNN Experiment Results and Analysis

The crack data images used in this experiment have been expanded by CE-GAN with a total of 6500 images, including 759 original images from the inspection results of a power station dam project. Each image in this dataset is of 256 × 256 resolution. The dam crack dataset was labeled and partitioned following the VOC2007 dataset format, of which 5000 sheets were used for training and 1500 sheets for testing.

We selected the SSD [28], YOLOv3 [20], Faster-RCNN [15], TDD-Net [18], DDN [17], and AF-RCNN for our experiments and compared them in terms of mAP, average IOU, and AR, respectively. The SSD, YOLOv3, and Faster-RCNN are the original aspect ratio and the scale size.

We counted the original 759 dam crack images and obtained a total number of 1270 cracks. We observed that the dam cracks are mainly straight, with relatively few reticulated cracks, and the reticulated cracks that exist can be decomposed into multiple straight cracks. Therefore, we counted the aspect ratio of linear cracks and designed an anchor size ratio suitable for the cracks of the dam.

We used the method in Section 3.2.2 to count the aspect ratio of cracks and improve the aspect ratio of the basic anchor based on the counted crack aspect ratio. The statistical graph of the aspect ratio of the cracks in the dam is shown in Figure 14. Finally, we designed the anchor size scale suitable for the dam cracks. We found that the aspect ratios of dam cracks can be classified into four types, which are 4:1, 2:1, 5:1, and 1:2. Based on the statistical results and the percentage of the four aspect ratios of the dam cracks, we adjusted the ratio of the basic anchor to (2:1, 4:1, 1:2, 5:1). In order to be able to detect cracks of smaller size and reduce the leakage cases, we will adjust the scale of the anchor to $\{32^2, 64^2, 128^2\}$, so that the number of anchors on the feature map is 12.

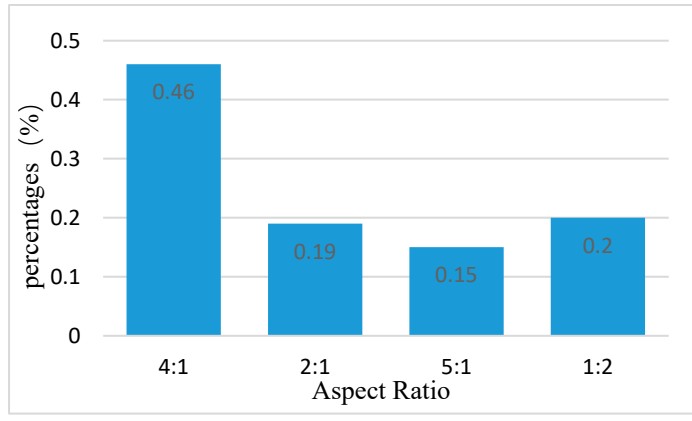

**Figure 14.** The graph of crack aspect ratio.

In summary, the anchor point aspect ratio of AF-RCNN is $(2:1, 4:1, 1:2, 5:1)$ and the scale is $\{32^2, 64^2, 128^2\}$. The anchor point aspect ratio of TDD-Net is $(1:2, 1:1, 2:1)$ and the scale is $\{15^2, 25^2, 40^2, 60^2, 80^2\}$. The anchor point aspect ratio of DDN is $(1:2, 1:1, 2:1)$

and the scale is $\{64^2, 128^2, 256^2, 512^2\}$. SSD, YOLOv3, and Faster-RCNN are all original aspect ratio and scale.

The test results are shown in Table 1.

**Table 1.** Detection results of each detection model.

| Method | mAP (%) | IoU (%) | AR (%) |
|---|---|---|---|
| SSD | 64.01 | 78.31 | 67.85 |
| YOLOv3 | 65.65 | 75.26 | 68.29 |
| Faster-RCNN | 72.68 | 73.15 | 66.16 |
| TDD-Net | 80.08 | 81.20 | 69.54 |
| DDN | 78.26 | 79.65 | 68.67 |
| AF-RCNN (ours) | 81.07 | 80.56 | 70.13 |

We find that the detection accuracy of Faster-RCNN is better than that of SSD and YOLOv3 with essentially the same average IoU and AR, which indicates that the Faster-RCNN model is a better choice as the benchmark model. TDD-Net has the best performance in IOU, but it is not as good as AF-RCNN in mAP. The AF-RCNN has the highest mAP value, indicating that this model is more accurate in detecting the location of cracks in the dam compared with other detection models.

To further compare the enhancement effect of the model in this paper after improving the anchor ratio and adding the attention mechanism, we compare the Faster-RCNN, SENet + Faster-RCNN, Attention mechanism + Faster-RCNN, and AF-RCNN, respectively, for the experiments. The test results are shown in Table 2.

**Table 2.** Detection results of ablation experiment.

| Method | Anchor | mAP (%) | IoU (%) | AR (%) |
|---|---|---|---|---|
| Faster-RCNN | 9 | 72.68 | 73.15 | 66.16 |
| Faster-RCNN | 12 | 74.72 | 76.26 | 66.85 |
| SENet + Faster-RCNN | 12 | 76.65 | 79.68 | 68.29 |
| Attention mechanism + Faster-RCNN | 12 | 78.50 | 80.25 | 67.86 |
| AF-RCNN | 12 | 81.07 | 80.56 | 70.13 |

As shown in Table 2, the detection accuracy of the model improved from 72.68% to 74.72% when the anchor aspect ratio was improved. When the feature extraction module is embedded in SENet, it can improve the model feature representation and the detection rate is increased to 76.65%. When the fusion of candidate proposal boxes is achieved by using the attention mechanism, the detection accuracy of the model is improved to 78.50%. Combining the three improvements at the same time, the model achieved a detection rate of 81.07%. In summary, the improvement of this model is effective.

As shown in Figure 15, the detection results of the model before improvement had obvious misses and the target proposal box was not accurate enough for locating the target cracks. However, the detection results of the improved model can be seen to be more accurate in locating the target cracks and reducing the number of wrong detections. We

can assume that the present model can effectively contribute to the improvement of the detection accuracy.

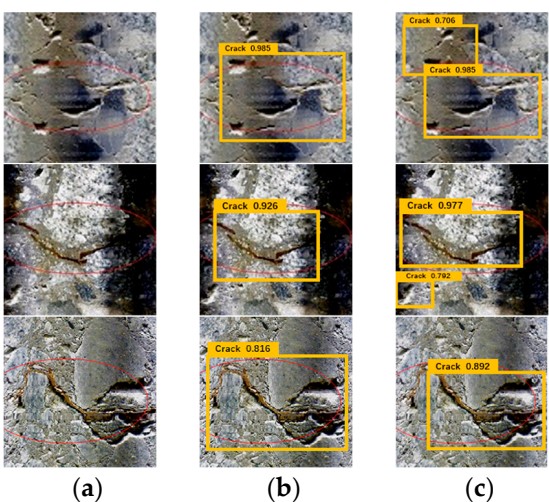

**Figure 15.** Detection effect comparison diagram. (**a**) Original image of cracks. (**b**) Test results before improvement. (**c**) Test results after improvement.

## 5. Discussion

On the basis of the existing research at home and abroad, we deeply analyzed the characteristics of dam crack images, used the improved confrontation generation network to expand the dam crack image set with high quality, and studied the dam crack image detection model based on deep learning to improve the accuracy of dam crack detection. Although our experimental results show that the recognition accuracy of our proposed model can reach 81.07%, there is still much space for improvement.

(1) In view of the low computational efficiency of existing deep learning methods in dam crack detection, we assume the use of transfer learning for target detection. Transfer learning can also alleviate to some extent the limitations caused by insufficient data on dam cracks. For example, Fan et al. [29] realized knowledge transfer of crack image features using a multi-level adversarial transfer network. Yang et al. [30] proposed an automatic pixel-level crack detection method based on deep transfer learning. Maybe we can try to apply transfer learning in dam crack detection.

(2) Existing crack detection methods still suffer from background noise interference, such as dirt patches and pitting. In this paper, we analyze low-pass filtering and high-pass filtering to design a filter suitable for dam crack images, so as to achieve feature enhancement for dam cracks. We feel that perhaps different scale information and different field of view information can be used to better recognize dam cracks. Zhang et al. [31] designed an encoder–decoder crack segmentation network based on multi-scale contextual information enhancement to make the network more effective at distinguishing between cracks and background noise. Qu et al. [32] proposed a new multi-scale feature fusion module where the deep semantic information is integrated into the low-level convolution stage layer by layer to strengthen the network model's ability to locate the crack pixels.

## 6. Conclusions

In this paper, we used an improved adversarial generative network to achieve a high-quality expansion of the dam crack image set for the characteristics of dam crack images and studied the dam crack images based on a deep learning model to solve the problem of lack of data volume while improving the accuracy of detection. We proposed a generative adversarial network (CE-GAN) based on crack feature enhancement to augment the dataset, which has been experimentally verified to improve the quality of the generated

images and content with the demand of expanding data samples. In addition, we also proposed an attention mechanism-based crack image detection model (AF-RCNN), which is improved in three parts: feature extraction part, candidate boxes detection region, and detection processing part. It is experimentally verified that the model improves the accuracy of locating the crack location of the dam and shows advantages in detection accuracy. In future research, we may apply other deep learning models in the dam crack detection to further improve the detection efficiency and accuracy.

**Author Contributions:** Conceptualization, G.X. and X.H.; methodology, G.X.; software, X.H.; validation, G.X., X.H., and Y.Z.; formal analysis, X.H.; investigation, X.H.; resources, X.H.; data curation, Y.Z.; writing—original draft preparation, Y.Z.; writing—review and editing, Y.Z.; visualization, C.W.; supervision, C.W.; project administration, G.X.; funding acquisition, G.X. All authors have read and agreed to the published version of the manuscript.

**Funding:** This research is partially supported by the National Key R & D Program of China under Grant No. 2018YFC0407106.

**Data Availability Statement:** Data generated and analyzed during this study are available from the corresponding author by request.

**Conflicts of Interest:** The authors declare no conflict of interest.

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
