# Peer review of "Dam Crack Image Detection Model on Feature Enhancement and Attention Mechanism"

_water, doi:10.3390/w15010064_

Round 1

Reviewer 1 Report

I have reviewed this MS these days. Generally, I feel that this MS is well-written and authoritative. It is a high-quality paper, deserving of publication in Water. However, I also spotted some problems in its current version, while some issues should be better explained. In sense, in order to make the MS more effective, I recommend a reconsideration of this MS after a major revision. Please see my detailed comments below.

(1)  Line 39-42. The authors mentioned two major problems for dam crack detection here. It is too early to mention this because no current studies on this aspect are reviewed. This problem is with respect to the structure of this MS, as the authors divide the introduction and related works separately into different sections. I suggest the author combine section 1 and 2 together, by highlighting the current studies before they mention the existing problems.

(2)  Line 114-115. The authors divide the current approaches into two kinds, the one-stage series and two-stage series. I don’t quite agree with the authors because as I think the one-stage algorithm is focusing on detection, and the two-stage algorithm should be much closer to recognition or segmentation. They are oriented from two different purposes, and ideally, recognition is sometimes based on detection.

(3)  Line 120-139. There are many studies regarding crack detection using DL methods. However, a limited number of these studies are reviewed. Some references are suggested, in which the current studies of DL for crack detecting are well reviewed.

[1] Mohammed MA., et al., Exploring the Detection Accuracy of Concrete Cracks Using Various CNN Models, Advances in Materials Science and Engineering, 2021.
[2] Han Z., et al., Vision-Based Crack Detection of Asphalt Pavement Using Deep Convolutional Neural Network, Iranian Journal of Science and Technology, Transactions of Civil Engineering, 2021

(4)  Line 143. Figure 1 is essential in the MS. I suggest the authors add some more details about the flowchart of their model.

(5)  Line 182. “The crack profiles are mostly linear”. It is difficult to understand how a crack profile is linear. Do the authors mean by “straight” instead of “linear”?

(6)  Line 183, and Line 192. In this subsection, the main features of the crack images are summarized. However, as to the “the crack images are not clear enough”, and “the dam crack samples set is lacking”. These two features relate to the source images in the dataset. They are not common features of the crack images.

(7)  Figure 4 in Line 224. Here the crack images are gray images while the ones in Figure 3 are color images. Are they from the same source?

(8)  Section 3.2.2, from Line 313-351. The authors have provided a very detailed description of the method of how they measure the length and width of the crack. However, this part is not described at all in the following experiment analysis. I suggest the author add measurement results in the experiment section.

(9)  Line 391-398. I am quite confused about their dataset of the crack images.  In Line 392, the author mentioned that some open-source crack dataset was assembled, but in Line 395, they said most of the images are from a power station dam. Again, in the data availability statement at the end of the MS, the PASCAL VOC2007 is cited. The authors should provide exact information about the image sources. The full datasets, rather than the partial ones, should be provided in the data availability statement.

(10)           In the abstract, the authors mentioned that the mAP value of the proposed model is approximately 81%, which is higher than other current methods such as SSD, YOLOv3. However, the TDD-net has a very close performance to the AF-RCNN, even it has a better IoU compared to AF-RCNN.

(11) In Table 2, the authors mentioned that model can be enhanced after adding an attention mechanism. If the attention mechanism is added here, I am confused about what the AF-RCNN means in Table 1, because AF-RCNN has been defined in prior in the abstract. I also suggest the authors check the data in these two tables, as the data in these two tables seem to conflict with each other. Such as the IoU value of Fast-RCNN reduces from 78.25% in Table 1 to 75.76% in Table 2.

(12) Discussion section is missing! I think there are many issues deserving to discuss in the MS.

Author Response

Dear reviewers:

We want to thank reviewer for constructive and insightful criticism and advice about the reference. Those comments are valuable and very helpful. We have read through comments carefully and have made corrections. Based on the instructions provided in your letter, we uploaded the file of the revised manuscript.

We will give a brief reply to your comments, and please read our manuscript for specific modifications.

Reviewer #1:

(1) We made appropriate changes to the contents of the first and second sections, adding the description of the relevant background. However, we didn't combine the first section and the second section. We thought these two sections still played different roles. We listened to the comments of the reviewers, and we explained the present situation to some extent before we mention the existing problems.

(2) We agreed with the reviewer's point of view, and delete the explanations about the one-stage and two-stage.

(3) We searched for papers about the application of deep learning in crack detection and added them to our manuscript.

(4) We explained Figure 1 in more detail.

(5) We changed “linear” to “straight”.

(6) We present a unified analysis of the problems of dam fracture images and the problems of dam fracture image datasets, so that the reader can more precisely understand our ideas

(7) Figure 4 and Figure 3 are the same source. In order to avoid this kind of misunderstanding, we added the original picture of Figure 4 in Figure 4. Figure 4 is an effect image produced by filtering the Figure 3.

(8) We added the presentation of the results of section 3.2.2 in the experimental part.

(9) We revised the description of the data source. We didn't use the VOC 2007 data set, which was a clerical error. The crack dataset was assembled into a database from the results of a dam project inspection at a power station.

(10) We made a supplementary analysis of the experimental results of TDD-Net.

(11) We admit that this was a mistake when we wrote the manuscript. To avoid this error, we re-run the code which uses the Faster-RCNN model and update the data. What puzzles you is our description of the network model used in the experiment. Table 2 is the result data of ablation experiment. we compare the Faster-RCNN, SENet + Faster-RCNN, Attention mechanism + Faster-RCNN and AF-RCNN respectively for the experiments.AF-RCNN includes SENet and Attention mechanism.

(12) We added the discussion section.

We would love to thank you for allowing us to resubmit a revised copy of the manuscript and we highly appreciate your time and consideration.

Sincerely.

Yuwei Zhang.

Reviewer 2 Report

The paper is well-written and well-organized. I only have some fundamental questions. Based upon the images provided in the paper, the cracks can be seen clearly by visual inspection.  Once the crack can be seen and deteced by eye balls, why do you need to use deep learning method to detect cracks? Please justify the necessity to use deep learning method here. Furthermore, the accuracy from the proposed method is over 80%, but the visual inpsect can easily identify the cracks (I think accuracy can reach 90%, based upon the images in the paper). Then, what is your contribution on crack detection?   

Author Response

We want to thank reviewer for constructive and insightful criticism and advice about the reference. Those comments are valuable and very helpful. We have read through comments carefully and have made corrections. Based on the instructions provided in your letter, we uploaded the file of the revised manuscript.

Reviewer #2:

(1) The first problem is that the introduction needs to provide sufficient background and include all relevant references.

On the basis of the first draft, we added relevant background descriptions and references.

(2) The second problem is that  why do you need to use deep learning method to detect cracks.

Many dams in China are built in complex terrain areas, and there is no way to manually inspect each dam due to the local human resources. The starting point of our research is to identify cracks through deep learning. If it can be applied to actual landing projects, it is only necessary to take regular long-distance full-coverage photographs of the dam surface of the reservoir, and the computer can automatically analyze the data carefully, quickly discover tiny cracks on the dam surface and slight changes in the dam structure, and realize automatic early warning and prediction.

(3) The third problem is that the accuracy about results.

We admit that the accuracy of the results did not meet our expectations. We think that there are many reasons that lead to the fact that the accuracy of the results is not as high as expected. Maybe it's the training time or the choice of network model. But we suppose the main reason is the data set. Our images of dam cracks are not as clear as those shown in the manuscript. Because these images are all from large-scale photographs, some tiny cracks may not be clearly identifiable. In addition, the number of data sets also limits the further training of the model. In the future research, we will expand the data set and improve the crack feature enhancement module to improve the accuracy.

(4) The forth problem is that our contribution on crack detection.

We provide a dam crack image detection model based on crack feature enhancement and attention mechanism. Firstly, we expand the dam crack image dataset through a generative adversarial network based on crack feature enhancement (Cracks Enhancements GAN, CE-GAN). It can fully expand the dam crack data samples and improve the quality of the training data. Secondly, we propose a crack image detection model based on the attention mechanism (Attention based Faster-RCNN, AF-RCNN). The attention mechanism is added in the crack detection module to give different weights to the proposal boxes around the crack target and fuse the candidate boxes with high weights to accurately detect the crack target location. We mainly provide some reference ideas for other researchers in the aspect of crack feature enhancement and the application of attention mechanism.

We would love to thank you for allowing us to resubmit a revised copy of the manuscript and we highly appreciate your time and consideration.

Round 2

Reviewer 1 Report

I appreciate that the authors have addressed most of my comments arisen in the first reviewing round. The quality of this MS has been improved and could be recommended for publication in WATER once the following comments are addressed.

(1)  As I noticed in the previous reviewing stage, Section 3.2.2 are the algorithm designed for extracting the geometric of the cracks, which majorly focuses on the length and width of two types of cracks. Although the authors responded that they have added the presentation of the results of section 3.2.2 in the experimental part, I failed to find the added description of this content. I suggest, again, the authors should delete this section if the function of length and width extraction lacks result support.

(2)  Line 469. The authors added that “the method in Section 3.3.3 to count the aspect…”. I could not be able to find the Section 3.3.3 in the revised MS. Please carefully check this sentence.

(3)  Line 417-420. The statement of the image dataset remains questionable and confusing, as well as the authors’ responses. Please CLEARLY describe the source of the dataset. 

Author Response

Thanks again for the review's correction!

Reviewer 2 Report

The authors need to revise their algorithm to improve the accuracy.  If the visual inpsect can easily identify the cracks, I don't know why the deep learning method can not identfy them. 

Author Response

Thanks again to the reviewers for reading!

Round 3

Reviewer 2 Report

I have no further comment.